# Efficiency Evaluation of China’s Public Sports Services: A Three-Stage DEA Model

**DOI:** 10.3390/ijerph182010597

**Published:** 2021-10-10

**Authors:** Pengyu Ren, Zhaoxia Liu

**Affiliations:** 1College of Architecture and Urban Planning, Chongqing Jiaotong University, Chongqing 400074, China; pengyu_ren@cqu.edu.cn; 2College of Architecture and Urban Planning, Chongqing University, Chongqing 400044, China; 3College of Physical Education, Chongqing University, Chongqing 400044, China

**Keywords:** public sports service efficiency, three-stage DEA model, technical efficiency, pure technical efficiency, scale efficiency, data envelopment analysis, China

## Abstract

Improving the level of public sports services enhances citizens’ physical fitness by implementing the national fitness program. A systematic and scientific efficiency evaluation is a prerequisite for optimizing and improving the level of public sports services in China. Based on data of the Chinese Statistic Yearbook, this study adopted the three-stage data envelopment analysis (DEA) model to measure and analyze the efficiency of public sports services in 31 provinces in China in 2016. To analyze the efficiency of public sports services, technical efficiency was decomposed into pure technical efficiency and scale efficiency. Simultaneously, environmental variables were added to improve accuracy. The results showed that scale efficiency was overestimated, and external technical efficiency was underestimated, before the elimination of external factors and environmental variables. Environmental factors significantly impacted the efficiency of public sports services. Regional gross domestic product (GDP) had a potentially positive impact, while population size partially restricted public sports service efficiency. After eliminating the impact of environmental and random factors, the comprehensive efficiency, pure technical efficiency, and scale efficiency of public sports services all showed improvement in varying degrees. The results provide beneficial insights for the formulation of rational improvement policies for public sports services.

## 1. Introduction

With rapid economic development and improved quality of life in China, physical exercise has become one of the most important activities in people’s daily life. Regular physical activity improves physical health, reduces negative psychological conditions such as anxiety and depression, and promotes self-expression in socialization [1]. The Chinese government has increasingly emphasized the importance of public participation in sports [2], especially in the context of the national “sport for all” strategy and the goal of building a modern socialist country that leads in sports [3,4]. The improvement of public sports services, as a fundamental part of the public service system, has become an important tool to implement national strategies and highlight the comprehensive competitiveness of sports in China. The creation of a more vibrant sports culture that meets public demand for diversified sports has become an increasingly prominent topic. The current academic consensus is that the uneven and inefficient distribution of public sports services in China stems from insufficient supply [5], specifically, the number of public sports venues per 10,000 people in Jiangsu was approximately 3.5 times that of Shanxi. Sports funding constraints and incomprehensive sports awareness of magistrates make it difficult for the Chinese government to invest significantly in public sports services in the short term [6]. For example, the per capita government expenditure on public sport services of Beijing was CNY 912, which was obviously higher than the less developed provinces (Heilongjiang: CNY 140; Anhui: CNY 135; Guizhou: CNY 189). Conversely, residents’ diverse sports needs exceed the current resources allocated to public sports services [7]. Therefore, given the limited public sports service investment, improving and optimizing China’s public sports service system can play an important role in improving the efficiency of public sports services and citizens’ quality of life.

Systematically and scientifically evaluating the efficiency of public sports services is a prerequisite for optimizing Chinese public sports services. Data envelopment analysis (DEA) models have been widely used in public service efficiency evaluation. Zhang and Pan used a DEA model to measure public health service efficiency and make recommendations based on panel data for 32 township health centers in rural Beijing, from 2007 to 2009 [8]. Shang et al. used a DEA model to measure public environment service efficiency for 30 Chinese provinces from 2006 to 2016 [9]. Combining a DEA model with geographic information system (GIS) technology, Wei et al. incorporated the operational efficiency and access equity of public transport services within a comprehensive framework to comprehensively assess public transport service performance [10]. Deviating from the classic DEA model, Tapia et al. proposed a bootstrap method to establish a confidence interval for the estimation of the population DEA efficiency score vector for a set of public service-producing units [11]. By measuring the efficiency of public service organizations, Smith and Street compared two tools (DEA and stochastic frontier analysis (SFA)), discussed specific model construction issues, and suggested optimizations [12]. Cui and Su proposed a series of specific initiatives for public sports services to guarantee equal access for all by studying the current situation of equality in public sports services in the Hebei Province [13]. Niu and Zhang’s study showed that while scale efficiency is relatively high for most colleges and universities, the proportion of public sports service facilities is a considerable problem [14]. Makubuya and Wang used a DEA model to measure and analyze the efficiency of public sports services in prefecture-level cities in the Zhejiang Province from 2008 to 2012 and demonstrated that the average efficiency level of public sports services in China is gradually increasing, while the average efficiency differential between regions is gradually decreasing; based on these findings, they further explored the factors influencing the efficiency of public sports services level through a Tobit model [15].

While the DEA model has been widely used in literature on the efficiency of public sports services, it does not exclude the influence of environmental and random error factors on efficiency. Therefore, the results obtained may have large deviations due to time and geographical differences. Besides, few studies have explored the factors affecting the efficiency of public sports services, and the lack of a uniform standard makes it difficult to reach a consensus for effective government policymaking.

This study addressed the shortcomings of the traditional one- and two-stage DEA methods by adopting the nonparametric three-stage DEA approach. The latter has been widely used to assess the efficiency of various industries by combining the advantages of both DEA and SFA methods and eliminating the effects of external environmental factors and statistical noise for more accurate efficiency evaluation [16]. Lee examined the impact of environmental factors and statistical noise on the efficiency of 89 global forestry and paper companies in 2001, using a three-stage DEA model, and demonstrated that environmental factors affected efficiency scores [17]. Fuentes et al. used a three-stage DEA approach to assess the learning–teaching process in higher education for technical efficiency and showed that the method improved the quality of results compared with those applied in previous studies [18]. Compared with traditional DEA, three-stage DEA captures potentially relevant exogenous variables, which may be important for efficiency results. Additionally, Zhang et al. used a three-stage DEA modeling approach to eliminate the interference effect of the external environment and statistical noise on industrial eco-efficiency measures and quantified the actual industrial eco-efficiency values of 30 Chinese provinces (autonomous regions and municipalities) from 2005 to 2013 [19]. Li and Lin used a three-stage data envelopment model to analyze the impact of government measures on the green productivity growth of China’s manufacturing industry during the Eleventh Five-Year Plan period (2006–2010) [20]. Zeng et al. used a three-stage DEA model to study and make policy recommendations about investment efficiency in China’s cultural industry [21]. Xu et al. examined social media’s impact on the operational performance of national tourist attractions using a three-stage DEA model [22]. Chen et al. applied a three-stage DEA procedure to comprehensively assess the operational efficiency of public hospitals in China from 2011 to 2018, showing that the operational environment is critical to public hospitals’ operational efficiency [23]. Most evaluations and studies have focused on the energy efficiency, eco-efficiency, and economic efficiency of other regional industries; applications in the field of public sports services have seldom been considered.

The present study measured and evaluated the efficiency of public sports services in 31 Chinese provinces by combining relevant data from China’s statistical yearbooks. It also provides recommendations for the scientific reorientation of public sports service investment for a rational optimal allocation plan that promotes public sports service efficiency.

## 2. Materials and Methods

To simultaneously eliminate the effects of external environmental factors and statistical noise on the efficiency estimates, Fried et al. combined the DEA and SFA models and proposed a three-stage DEA model [24].

### 2.1. Stage 1: Traditional DEA Model

The traditional Charnes–Cooper–Rhodes (CCR) model was developed by Charnes et al. [25]. However, the CCR input–output analysis for calculating efficiency values assumed constant returns to scale, which does not apply to the situation of variable return to scale [26]. To address this inconsistency, Banker et al. proposed the DEA–BCC model that assumes variable returns to scale and decomposed the technical efficiency (TE) into pure technical efficiency (PTE) and scale efficiency (SE) under the assumption of constant returns to scale (TE = PTEXSE) [27]. Compared with the technical efficiency under the CCR model, the input indicators selected in the variable returns-to-scale BCC model for measuring and evaluating the efficiency of public sports services are easier to control than the output indicators are. The input-oriented BCC model, under the assumption of variable returns to scale, is more mature and eliminates the influence of scale factors, thereby accurately reflecting the operation and management level of the evaluated decision-making unit (DMU). Therefore, the BCC model characterized by input orientation was chosen in Stage 1 of the present study to produce more reasonable results.

Equation (1) was used to calculate each DMU’s efficiency value for a total of n DMUs, each with m inputs and s outputs.
(1)$$\begin{array}{l} \min\;\theta \\ s.t\;\;\;\sum_{i=1}^{n}X_{i}\lambda_{i}+s^{-}=\theta X_{0}\\ \;\;\;\sum_{i=1}^{n}Y_{i}\lambda_{i}-s^{+}=Y_{0}\\ \;\;\sum_{i=1}^{n}\lambda_{i}=1\\ \;\;\lambda_{i}\geq 0,\;i=1,2,\ldots ,n\\ \;\;s^{+}\geq 0,\;s^{-}\geq 0 \end{array}$$
where $X_{i}={\left(x_{i1},x_{i2},\ldots ,x_{im}\right)}^{T}$ is the input vector of the DMU *i*; $Y_{i}={\left(y_{i1},y_{i2},\ldots ,y_{is}\right)}^{T}$ is the output vector of the DMU *i*; $\lambda_{i}$ are the weights of the DMU *i*; $s^{-}$ indicates the amount of input slack adjustment; $s^{+}$ indicates the amount of output slack adjustment; and θ is the pure technical efficiency values, where 0≤θ≤1 and the pure technical efficiency increases as θ approaches 1.

### 2.2. Stage 2: SFA Model

The first-stage input–output slack variables are influenced by external environmental factors, random errors, and internal management factors. The traditional DEA model does not accurately reflect which of these affects the efficiency values, attributing the entire effect to internal management. Each DMU’s efficiency and input difference values were calculated in Stage 1, wherein the input difference value was the difference between the DMU’s actual and optimal efficiency inputs. To remove external environmental and random error effects on the efficiency estimates, the SFA model was used to decompose the input difference values. Given p environmental variables, the SFA regression equation was constructed for m input slack variables of n DMUs, according to Battese and Coelli’s methodology, as follows [28]:(2)$$\begin{array}{l} s_{ki}=f^{k}(z_{i};\beta^{k})+\upsilon_{ki}+u_{ki}\\ i=1,2,\ldots ,n;\;k=1,2,\ldots ,m \end{array}$$
where $s_{ki}$ is the *k*th input slack of the *i*th DMU; $z_{i}{=(z}_{1i}{,z}_{2i},\ldots {,z}_{pi})$ indicates the environment variable; $\beta^{k}$ represents the environmental variable parameter to be estimated; $f^{k}(z_{i};\beta^{k})$ denotes the effect of environmental variables on the input differential $s_{ki}$, which is represented as $f^{k}(z_{i};\beta^{k})=z_{i}\beta^{i}$; $\upsilon_{ki}+u_{ki}$ is the mixed error term, where $\upsilon_{ki}$ is the random disturbance, assuming $\upsilon_{ki}\sim N(0,\delta_{k\upsilon }^{2})$; $u_{ki}$ represents inefficient management, assuming that $u_{ki}$ obeys a truncated normal distribution, that is, $u_{ki}\sim N^{+}(u^{k},\delta_{u\upsilon }^{2})$, and $\upsilon_{ki}$ is independently uncorrelated with $u_{ki}$. Specifically, the influence of the management factor dominates when $\gamma =\frac{\delta_{uk}^{2}}{\delta_{uk}^{2}+\delta_{\upsilon k}^{2}}$ approaches 1. When $\gamma =\frac{\delta_{uk}^{2}}{\delta_{uk}^{2}+\delta_{\upsilon k}^{2}}$ approaches 0, the random error effect dominates.

The SFA regression results were then used to adjust the input term of each DMU and increase the input for DMUs that were located in a better environment, to both eliminate the influence of environmental and random factors and measure the efficiency value purely reflecting each DMU’s management level. The input of the most efficient DMU was used as a benchmark, and the input for every other DMU was adjusted as follows:(3)$$\begin{array}{l} x_{ki}^{*}=x_{ki}+[\max_{i}\{z_{i}\overset{\wedge }{\beta^{k}}\}-z_{i}\overset{\wedge }{\beta^{k}}]+[\max_{i}\{\overset{\wedge }{\upsilon_{ki}}\}-\overset{\wedge }{\upsilon_{ki}}]\\ i=1,2,\ldots ,n;\;k=1,2,\ldots ,m \end{array}$$
where $x_{ki}$ and $x_{ki}^{*}$ denote the inputs to the DMU before and after the adjustment, respectively; $\overset{\wedge }{\beta^{k}}$ denotes the estimates of the environmental variables; and $\overset{\wedge }{\upsilon_{ki}}$ denotes the estimates of the random disturbance terms. The first bracketed term indicates the adjustment of all DMUs to the same environment. The second bracketed term indicates the adjustment of all DMUs’ random errors to the same situation, equalizing each DMU’s external environment and luck factors.

### 2.3. Stage 3: Adjusted BCC Model

The third stage of the DEA model involved the substitution of the original output and input index data of all DMUs adjusted in the previous stage into the first stage, that is, the BCC model, to measure the efficiency values of all DMUs.

## 3. Data and Variables

### 3.1. Selection of Input and Output Indicators

The reasonable determination of input and output indicators is a prerequisite for evaluating the efficiency of public sports services through the DEA model. Input and output indicator selection for public sports services is a complex process, covering human, financial, and material factors. Most existing studies in this sphere are qualitative. In the present study, the annual per capita public sports expenditure and the proportion of public sports expenditure in local fiscal expenditure were selected as the input indicators based on Zhang’s analysis [29]. This is because the resources allocated by each local government to public sports services involve human resources, money, and materials and are ultimately reflected in the funding. Five output indicators were selected—the number of public sports venues per 10,000 people, the area of public sports venues per 10,000 people, the number of sports guidance stations per 10,000 people, the number of social sports instructors per 10,000 people, and the number of national physical fitness monitoring stations per 10,000 people [30]. The amount of public sports services resources received can represent the efficiency and magnitude of public sports services resources controlled by local governments.

### 3.2. Selection of Environment Variables

Environmental variables refer to factors that can affect the efficiency of public sport services and are beyond subjective control. Considering the factors that affect the efficiency of public sports services [31,32,33], this study used the regional gross domestic product (GDP) and population size as environmental variables. Regional GDP, a core national economic accounting indicator, captures a region’s economic status and development. The level of affluence of a place positively affects the efficiency of its government, which provides better and more efficient public sports services [34]. The regional GDP allows local governments to allocate production resources, provide public sports goods and services, and improve the efficiency of public sports services. Furthermore, each region’s population is based on the natural environment and economic activities, and any rapid change in population size greatly impacts the region’s social resources and environmental development, which directly affects the efficiency of government public sports services [35]. Other scholars have argued that the population density of a region is negatively correlated with the cost of government management and supervision. In this case, increasing population size has a "scale effect" on the public sports services provided by local governments, leading to lower average costs and higher efficiency of public sports services provided by the government.

### 3.3. Data Sources

The basic data on the input and output indicators used in this study were obtained from the China Sports Industry Statistical Yearbooks and China Statistical Yearbooks. Missing data for individual provinces were filled using moving averages.

## 4. Results and Discussion

### 4.1. Results of the First-Stage DEA

An input-oriented BCC model was applied in the first stage, and the DEA software DEAP 2.1 was used to calculate the TE, PTE, and SE of each province.

As shown in Figure 1 (values of Figure 1 are shown in Table A1), the average TE, average PTE, and average SE values for all provinces in China were 0.557, 0.688, and 0.902, respectively, excluding the influence of external environmental factors and random noise errors. This implies that the comprehensive efficiency of public sports services relies on existing technology, and investment can be improved by 44.3% before it reaches the government production frontier; the SE for most provinces is significantly greater than the PTE, which indicates that the TE of public sports services mainly arises from the PTE, rather than the SE. These results indicate that PTE is the main factor restricting the efficiency of public sports services. However, in the first stage of DEA, which does not consider external environmental and random noise error factors, it cannot be definitively concluded whether the PTE and SE have been underestimated and overestimated, respectively.

### 4.2. Results of the Second-Stage SFA

The SFA model was used to analyze the input slack variables obtained in the first stage, and environmental variables such as regional GDP and population size were taken as independent variables. The results were substituted into the SFA linear regression model. With the help of the FRONTIER4.1 software, the maximum likelihood method was used for SFA regression analysis to analyze whether the external environmental variables significantly affected the difference between the ideal and actual input variable values. Given such an impact, the external factors would be eliminated through Equation (3) to obtain the input variable X *. The results of the second stage are shown in Table 1.

Table 1 shows that for the two environmental variables and the input slack variables, per capita public sports expenditure and the proportion of public sports expenditure in local fiscal expenditure, the likelihood ratio (LR) test value is significant at the 5% level, indicating model reliability. Furthermore, at the 1% level of significance, the γ values of the two input slack variables tend to 1, indicating that environmental factors significantly impact input redundancy. Therefore, the external environment variables and random factors were eliminated using Equation (3) to ensure uniform external environment characteristics for all provinces, and to obtain accurate results in the third stage. The results for the environmental variables were as follows:

Regional GDP: The slack variable coefficients of the regional GDP variable to the per capita public sports expenditure and the proportion of public sports expenditure in local fiscal expenditure were negative, indicating that an increase in the GDP reduces the inputs of the abovementioned slack variables, which is conducive to TE improvement. A higher GDP means a better level of local economic development, which is conducive to improved living standards of residents [36]. Public desire for sports services is also likely to increase, generating scale effects. Among the two slack variables, only the per capita public sports expenditures passed the 1% significance test, unlike the proportion of public sports expenditure in local fiscal expenditure, demonstrating a key linkage between public sports efficiency and per capita public sports expenditure. It also reflects the significance of increasing per capita public sports expenditure.

Population size: The increase in population size negatively affects both input slack variables due to overcrowding. The T values of per capita public sports expenditure and the proportion of public sports expenditure in local fiscal expenditure were all positive and significant at the 1% level, indicating that the efficiency of public sports services did not increase with an increase in population size. On the contrary, population had a negative effect on the efficiency of such services.

### 4.3. Results of the Third-Stage DEA–BCC

After adjusting the public sports input variables for the 31 provinces, DEA–BCC calculations were repeated using DEAP 2.1 software. The results are shown in Figure 2 and the values are shown in Table A2 (Appendix A).

By comparing first- and third-stage results, it was found that the sports efficiency values for provincial governments changed greatly after eliminating the influence of environmental variables and random error factors, implying that the selected environmental variables affect public sports efficiency values. Compared with the first stage, the TE value in the third stage increased significantly, from 0.457 to 0.552, and the net TE and SE values increased by 15.31% and 3.24%, respectively. Furthermore, the SE of each province showed a sizable downward trend after the adjustment, indicating that diseconomies of scale are one of the reasons for the low efficiency of public sports services, rather than the result obtained in the first stage, which implies that the low efficiency of public sports services is due to pure technical inefficiency. The final analysis shows that each province’s SE was significantly overestimated, but the pure TE was significantly underestimated; as the degree of overestimation was higher than the degree of underestimation, the TE value was overestimated.

### 4.4. Provincial Analysis of Public Sports Service Efficiency in China

The third stage demonstrates the influence of environmental factors and random error factors, and the results truly reflect the efficiency of public sports services. Therefore, combined with the third-stage results, China’s actual public sports service efficiency situation was further analyzed.

The third-stage results indicated that the overall TE of public sports services in China was 0.675, which was not very high; the value of pure TE was 0.801, illustrating that the decision-making and management levels of most local governments in terms of public sports services are relatively mature. On the other hand, the low SE of public sports services can be attributed to the low level of comprehensive efficiency of public sports services in various provinces.

The comparison of the first-stage results showed that the efficiency of public sports services has changed greatly for local governments in China. The TE values of public sports services improved in 22 provinces, namely, Beijing, Hebei, Shanxi, Inner Mongolia, Liaoning, Jilin, Heilongjiang, Shanghai, Jiangsu, Zhejiang, Anhui, Jiangxi, Shandong, Henan, Hubei, Hunan, Guangdong, Chongqing, Sichuan, Yunnan, Shaanxi, and Gansu. Beijing, Shanghai, Liaoning, and Guangdong mainly benefited from scale efficiency; Shanxi, Jilin, Heilongjiang, Jiangsu, Zhejiang, Shandong, Henan, Hubei, Sichuan, and Yunnan benefited from scale and pure technical efficiency; Hebei, Inner Mongolia, Anhui, Jiangxi, Hunan, Chongqing, Shaanxi, and Gansu were limited by scale efficiency, but benefited from pure technical efficiency. The TE values of six provinces and cities declined, including Fujian, Guangxi, Guizhou, Qinghai, Ningxia, and Xinjiang; Ningxia, Tibet, and Hainan experienced a considerably sharp decline. In fact, their provincial governments were satisfied with the efficiency of sports in the first stage primarily because even without large sports input, their performance was outstanding due to local environmental factors or good luck. However, once these external variables were eliminated, the management levels and SE were not as pessimistic as expected. The decline in the PTE of public sports services in Guangxi and Qinghai was the main reason for the decrease in comprehensive efficiency. To summarize, whilst excluding environmental factors and random noise, the TE value was overestimated because of the changes in the PTE and SE of public sports services provided by local governments.

## 5. Conclusions

This study used the three-stage DEA model to systematically measure and analyze the efficiency of public sports services in China. The findings yielded the following conclusions.

(1) The TE, PTE, and SE of local governments’ public sports services experienced different degrees of change after using the SFA model to eliminate the influence of random errors and external environmental factors. This suggests that the difference in local environmental factors can have a significant impact on the efficiency of public sports services and should be eliminated for analytical accuracy. The PTE was underestimated and the SE was overestimated, indicating that local governments had a strong ability to allocate resources of public sports services, but did not have economies of scale overall and should maintain their current level of technical management and expand the scale of public sports industry development to realize the scale dividend.

(2) Regional economic development levels significantly impacted the efficiency of public sports services, as verified by the regression of the regional GDP on input redundancy in the second stage of the SFA regression analysis. This is in line with existing research findings that higher regional GDP is conducive to reducing the input of slack variables and will reduce input variable waste while increasing output. However, unlike existing studies, the slack variable of regional population on per capita public sports expenditure and the proportion of public sports expenditure in local fiscal expenditure was positive and significant (at the 1% level). This indicated that the expansion of the regional population size led to a waste of input variables, and reduced output, which indicated that appropriate control of the urban population was needed when developing the urban economy and improving the efficiency of public sports services.

(3) Apart from increasing returns to scale for public sports service efficiency in some provinces, the returns to scale for most provinces were found to be decreasing or constant. This suggested that some provinces may have had an unreasonable development of public sports services due to the limited regional development and the large-scale population, which indicated that the provinces need to increase investment to improve the efficiency of public sports services. However, the efficiency of public sports services in the provinces with constant returns to scale was at a best level.

From the perspective of increasing returns to scale, while increasing investment has a certain beneficial effect, blindly investing in public sports services may not necessarily produce better results, instead causing wastage and reducing efficiency.

Based on the empirical findings and the reality of the local governments’ public sports service situation, the following recommendations are proposed:

(1) The scale of public sports service investment should be determined reasonably, based on the returns to scale. For regions with increasing returns to scale, an increase in the input leads to higher output returns. The opposite holds true for diminishing returns to scale. Therefore, to improve the efficiency of public sports services in China, local governments should keep the local development situation in mind, to ensure that public sports services benefit more people.

(2) The technical progress of public sports services should be promoted vigorously. Technological progress is the main factor affecting the efficiency of public sports services in all provinces. At present, there is a lot of room to improve the technical efficiency of public sports services in many regions in China, to effectively meet the growing demand for public sports services through technical efficiency improvements. Therefore, to obtain the maximum output of public sports services for a given input, local governments must focus on improving technical progress, whilst considering technical efficiency.

(3) A comprehensive public service communication platform must be established to connect the government, social organizations, and residents, and to integrate various forces to support the provision of public sports services, such as through the participation of public and social organizations in decision-making. Sports are an important part of daily life; therefore, local governments should not only provide a variety of sports fitness alternatives, but also focus on the inputs and outputs of public sports service management, to improve the management efficiency, scale efficiency, and overall level of public sports services.

(4) The national physique monitoring mechanisms should be strengthened to promote the development of national fitness. National physique monitoring is an important method to inculcate the correct concept of fitness, execute scientific and comprehensive fitness regimens, and measure public sports development. Local governments’ research institutions can build a three-dimensional national fitness monitoring service module to increase publicity, popularize scientific fitness knowledge, and improve national fitness awareness. Furthermore, the purchase of health insurance cards for fitness services should be promoted, and the public should be encouraged to actively participate in national fitness and fitness monitoring activities.

Although some meaningful findings were yielded in this study, several gaps could be filled with future endeavors. First, this research only selected the cross-sectional data of each province in 2016 as the research object to calculate its service efficiency, the time factor can be taken into account in future research, and the public service efficiency of each province can be explored by analyzing the TE values of different years to explore the dynamic evolution of the development of local sports service levels, especially discussing the gap of the effectiveness of various influencing factors between different periods, such as usually scenarios and post-pandemic scenarios, to provide more accurate and comprehensive suggestions for policy implementation. In addition, to improve the practical value of this study, establishing an online database with geomarketing software to analyze the demand and supply of public sport services based on the method of the study is better to support urban planning of public support services, which is our future endeavors orientation.

## Figures and Tables

**Figure 1 ijerph-18-10597-f001:**
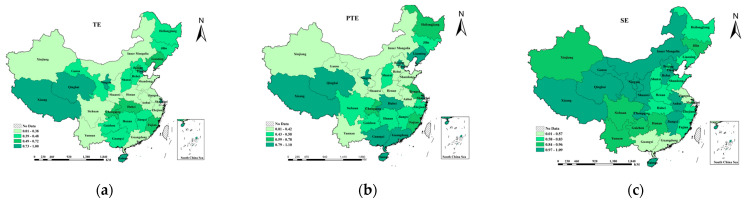
The first-stage technical efficiency and scale efficiency of provincial public sports service: (**a**) the technical efficiency of each province; (**b**) the pure technical efficiency of each province; (**c**) the scale efficiency of each province.

**Figure 2 ijerph-18-10597-f002:**
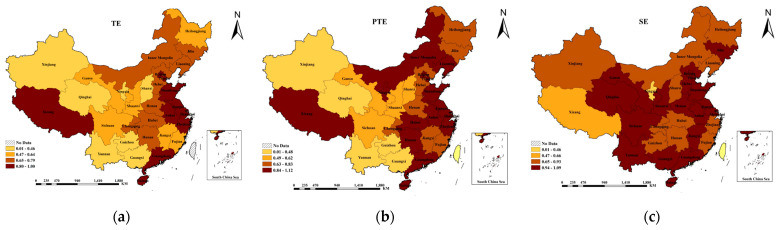
The adjusted technical efficiency and scale efficiency of provincial public sports service adjustment: (**a**) the technical efficiency of each province; (**b**) the pure technical efficiency of each province; (**c**) the scale efficiency of each province.

**Table 1 ijerph-18-10597-t001:** SFA model estimation results of input slack variables.

Variables	Per Capita Public Sports Funds			The Proportion of Public Sports Funds in Local Financial Expenditure		
	Parameters	St.d	t	Parameters	St.d	t
Constant	−3.023	4.542	−0.688	−17.550 ***	1.220	−14.566
β_1_	−0.262 ***	0.543	−0.623	−0.987 ***	0.218	−7.193
β_2_	−0.237 ***	0.496	−0.747	3.021 ***	0.219	11.646
σ^2^	0.738 **	0.315	2.322	2.195 **	0.312	5.724
γ	1.27 × 10^−4^ ***	4.25 × 10^−2^	3.431	0.9999 ***	0.000	5,031,724
Log likelihood	−27.782			−18.329		
One sided bias of likelihood ratio test	4.413 **			13.218 **		

Note: ***, ** represent the 1%, 5%, and 10% significance levels, respectively. T value is an index to test whether the explanatory variable has significant influence on the explained variable; σ^2^ represents the variance of the invalid rate item; γ is the ratio of the invalid rate term to mixed error term; LR means that the test value follows a mixed chi-squared distribution.

## Data Availability

Not applicable.

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
