# Peer review of "Efficiency Evaluation of China’s Public Sports Services: A Three-Stage DEA Model"

_ijerph, 2021, doi:10.3390/ijerph182010597_

Round 1

Reviewer 1 Report

Congratulations to the authors, it is a very extensive and applied good work. Its methods are well explained and it is also easy to read. Although the article is compact and its conclusions robust, it is possible to improve by incorporating a greater explanation of the economies of scale established in the conclusions and how these are articulated to improve the frontiers of efficiency.

Author Response

Many thanks for the reviewer’s reviews.

Q1:It is possible to detail this point more, especially that it includes the random effect in the results, (just to target a more general audience of readers).

A1:Authors have added specific sources of random effects and more details to make it more accessible to a general audience.

(lines 334-340): This suggests that the difference in local environmental factors can have a significant impact on the efficiency of public sports services and should be eliminated for analytical accuracy. The PTE was underestimated and the SE was overestimated, indicating that local governments had a strong ability to allocate resources of public sports services, but did not have economies of scale overall and should maintain their current level of technical management and expand the scale of public sports industry development to realize the scale dividend.

Q2: it may be interesting to delve into this point.

A2: Authors have added the specific reasons for the increasing returns to scale for public sports service efficiency in some provinces and the corresponding measures that should be taken are explored.

(lines 354-358): This suggested that some provinces may had an unreasonable development of public sports services due to the limited regional development and the large-scale population, that indicated the provinces need to increase investment to improve the efficiency of public sports services. However, the efficiency of public sports services in the provinces with constant returns to scale was at a best level.

Q3: Does it present any limitations associated with a new post-pandemic scenario.

A3: Discuss the gap between the usually scenario and post-pandemic scenario is interesting and meaningful. Therefore, we considered to select the idea to be the future research orientation. And added the limitation in lines (399-402).

(lines 399-402): especially discussing the gap of the effectiveness of various influencing factors be-tween different period, such as usually scenario and post-pandemic scenario, so as to provide more accurate and comprehensive suggestions for policy implementation.

Reviewer 2 Report

A good study. However, the quality can be improved by providing evidence of the problem and discussing the findings with reference to results from previous studies. 

The following specific comments should be attended to:

Abstract

More information could be provided on the methods, especially on the sampling technique used.

Introduction

In the first paragraph, the evidence of the problem, “insufficient supplyment” and “Sports funding constraints” should be provided. Use statistics if possible. The academic consensus is not sufficient evidence, especially since the topic has to do with improving quality of life, a practical (not theoretical) contribution. Therefore, provide evidence of the problem in reality.

“Supplyment” should be changed to “supply” throughout the document.

Materials and Methods

In 3.1, “Selection of input-output indicators” should be changed to “Selection of input and output indicators”

Results and Discussion

The results should be discussed in relation to existing literature. The studies should be compared and contrasted with other findings, similar or dissimilar, and the implications should be discussed.

Author Response

Many thanks for the suggestion.

Q1: More information could be provided on the methods, especially on the sampling technique used.

A1: Authors have added the data resource of this study in lines 11-12. and show the main method and study area. Because of long text of introducing the method, we describe the methods and data presented in the abstract, detailed method introduction can be seen in section the Section 3. Appreciate your understanding.

(lines 11-12): Based on data of Chinese Statistic Yearbook, this study adopted the three-stage data envelopment analysis (DEA) model to measure and analyze the efficiency of public sports services in 31 provinces in China in 2016.

Q2:In the first paragraph, the evidence of the problem, “insufficient supplyment” and “Sports funding constraints” should be provided. Use statistics if possible. The academic consensus is not sufficient evidence, especially since the topic has to do with improving quality of life, a practical (not theoretical) contribution. Therefore, provide evidence of the problem in reality.

A2: Authors has applied the practical statistic evidence to support the problem in reality “insufficient supplyment” and “Sports funding constraints” in lines 40-41 and lines 44-46, respectively.

(lines 40-41): specifically, the number of public sports venues per 10,000 people in Jiangsu was ap-proximately 3.5 times of Shanxi.

(lines 44-46): For example, the per capital government expenditure on public sport services of Beijing was 912 yuan, which was obviously higher than the less developed provinces (Hei-longjiang: 140 yuan, Anhui: 135 yuan; Guizhou: 189 yuan)

Q3:“Supplyment” should be changed to “supply” throughout the document.

A3: Authors have replaced “Supplyment” by “supply” in line 40. And checked the other part of manuscript to find the similar revision.

(line 40): The current academic consensus is that the uneven and inefficient distribution of public sports services in China stems from insufficient supply.

Q4:In 3.1, “Selection of input-output indicators” should be changed to “Selection of input and output indicators”.

A4: Authors replaced “Selection of input-output indicators” by “Selection of input and output indicators” in lines 179-183.

(lines 179-183):

3.1. Selection of input and output indicators

The reasonable determination of input and output indicators is a prerequisite for evaluating the efficiency of public sports services through the DEA model. Input and output indicator selection for public sports services is a complex process, covering human, financial, and material factors.

Q5: The results should be discussed in relation to existing literature. The studies should be compared and contrasted with other findings, similar or dissimilar, and the implications should be discussed.

A5: we added the gap between the existing literature and this study to discuss the similar or dissimilar, and the implications in lines 343-351.

(lines 343-351): This is in line with existing research findings that higher regional GDP is conducive to reducing the input of slack variables and will reduce input variable waste while increasing output. However, different from existing studies, the slack variable of regional population on per capita public sports expenditure and the proportion of public sports expenditure in local fiscal expenditure was positive and significant (at the 1% level). This indicated that the expansion of the regional population size led to a waste of input variables, and reduces output. That indicated appropriately control the urban population was needed when developing the urban economy and improving the efficiency of public sports ser-vices.

Reviewer 3 Report

OTHER an on-line database for collecting information relating to sports facilities (National Data Bank) IT SHOULD BE PROVIDED with geomarketing software to carry out pressure analysis of the demand for sports in relation to the supply of facilities, to support the planning of interventions (specify better) but also a citizen information portal for searching and viewing on a map of the nearest facilities where to practice the desired sport.

There is no indication of international regulations to have a reference not only Chinese but also international.

Author Response

Many thanks for the valuable suggestion

Q:OTHER an on-line database for collecting information relating to sports facilities (National Data Bank) IT SHOULD BE PROVIDED with geomarketing software to carry out pressure analysis of the demand for sports in relation to the supply of facilities, to support the planning of interventions (specify better) but also a citizen information portal for searching and viewing on a map of the nearest facilities where to practice the desired sport.

A: Providing an on-line database with geomarketing software is a great idea to improve the practical value of this study. Therefore, we added this in the limitation and future research orientation of this paper in lines 402-406. And devote to achieve this target in future plan.

(lines 402-406) In addition, to improve the practical value of this study, establishing an on-line database with geomarketing software to analyze the demand and supply of public sport services based on the method of the study is better to support urban planning of public support services, which is our future endeavors orientation.

Round 2

Reviewer 3 Report

The changes made have improved the text